# Which Low-Abundance Proteins are Present in the Human Milieu of Gamete/Embryo Maternal Interaction?

**DOI:** 10.3390/ijms20215305

**Published:** 2019-10-24

**Authors:** Analuce Canha-Gouveia, A. Paradela, António Ramos-Fernández, Maria Teresa Prieto-Sánchez, Maria Luisa Sánchez-Ferrer, Fernando Corrales, Pilar Coy

**Affiliations:** 1Department of Physiology, Faculty of Veterinary, University of Murcia, Campus Mare Nostrum, IMIB-Arrixaca, 30100 Murcia, Spain; analuce.canha@um.es; 2Proteomics Laboratory, Centro Nacional de Biotecnología, Consejo Superior de Investigaciones Científicas (CSIC), 28049 Madrid, Spainalberto.paradela@cnb.csic.es (A.P.); fcorrales@cnb.csic.es (F.C.); 3Department of Obstetrics & Gynecology, “Virgen de la Arrixaca” University Clinical Hospital, IMIB-Arrixaca, 30100 Murcia, Spain; mt.prieto@um.es (M.T.P.-S.); marisasanchez@um.es (M.L.S.-F.)

**Keywords:** low abundance proteins, human reproductive fluids, salpingectomy

## Abstract

The improvement of the embryo culture media is of high relevance due to its influence on successful implantation rates, pregnancy, neonatal outcomes, and potential effects in adult life. The ideal conditions for embryo development are those naturally occurring in the female reproductive tract, i.e., the oviductal and uterine fluids. To shed light on the differences between chemical and natural media, we performed the first comparative study of the low abundance proteins in plasma, uterine, and oviductal fluid collected, simultaneously, from healthy and fertile women that underwent a salpingectomy. The rationale for this design derives from the fact that high-abundant proteins in these fluids are usually those coming from blood serum and frequently mask the detection of low abundant proteins with a potentially significant role in specific processes related to the embryo–maternal interaction. The proteomic analysis by 1D-nano LC ESI-MSMS detected several proteins in higher amounts in oviductal fluid when compared to uterine and plasma samples (RL3, GSTA1, EZRI, DPYSL3, GARS, HSP90A). Such oviductal fluid proteins could be a target to improve fertilization rates and early embryo development if used in the culture media. In conclusion, this study presents a high-throughput analysis of female reproductive tract fluids and contributes to the knowledge of oviductal and uterine secretome.

## 1. Introduction

Assisted reproductive technologies (ART) have grown worldwide to assist the increasing number of patients who request these methods to conceive [1]. This successful development of ART has been achieved by continuous critical analysis of all methods performed and materials used [2]. Recently, culture media has received special attention [3]. During intracytoplasmic sperm injection (ICSI) or in vitro fertilization (IVF), the development of human-preimplantation embryos takes place in an artificial environment (i.e., a petri dish with chemically defined or semi defined culture media). Nowadays, it is possible to find culture media on the market composed of different formulations that vary from simple salt solutions to more complex compositions that include synthetically derived proteins and growth factors [4]. The presence of proteins, other than albumin in these chemically defined solutions, is usually avoided [5]. The batch-to-batch variability in protein composition has been linked to fluctuating pregnancy rates in clinics [6]. To overcome this variability, to minimize the risk of disease transmission and to achieve stable high success rates, protein supplement has evolved from initial donor or patient plasma/serum formulas to recombinant albumin [7]. Although current culture media are showing consistency and promote higher pregnancy rates than the ones used before, there is a growing concern of a possible link between the composition of the culture media and the phenotype of the offspring [8]. The developmental origins of health and disease (DOHaD) hypothesis proposed that in utero stress is associated with an increased risk of disorders in adulthood. This hypothesis has been extended to include the putative effect of the oviductal and periconceptional environment to future progeny health [9]. In animal models, extensive data revealed the presence of genetic and epigenetic alterations, especially in imprinted genes, derived from the use of ART [10,11,12]. In recent years, the long-term side effect of ART on further development in humans has also been studied. Several works have suggested that the composition of the media in which embryos are cultured may have an impact on the quality of embryos generated in IVF/ICSI cycles, thereby influencing implantation and pregnancy rates [8,13]. In 2016, a single-center cohort study did not find significant effects on birthweight, malformation risk (minor and major), offspring growth, and frequency of medical concerns according to culture media [14]. However, a contemporary study found a correlation between the IVF media and perinatal outcomes in a randomized controlled trial. The outcomes considered in this study were the number of viable embryos grown, the rate of successful implantations, the pregnancy rates, and the birthweights of the newborns [15]. Due to this uncertainty, it is imperative to invest more effort in the improvement of embryo culture conditions, trying to mimic the in vivo environment where embryos develop. Oviductal and uterine fluids (OF, UF) contain essential factors of different origin and composition that are pivotal for the development of gametes, zygotes, and later, embryos. These include nutrients, hormonal and non-hormonal factors, electrolytes, and other macromolecules [16]. Additionally, the volume, pH, and osmolality of these fluids are precisely regulated. If we compare reproductive fluids with artificial culture media, we observe that most of the lipids, hormones, glycosaminoglycans, proteins, and exosomes found in vivo are missed [16,17]. Even though embryos show some flexibility and are able to adapt to changes in growth-media, as in the case of protein-free media [18], it is clear that the multiple components of reproductive fluids play an active role in the whole process [19]. As aforementioned, human recombinant albumin is present in most IVF culture media. This protein has been recognized for its important role in embryo culture and is the most abundant macromolecule in the human oviduct [20]. This high proportion is a limiting factor to effective detection of low abundance proteins, which could play an important role for the embryo [21] and could be of interest in the design and development of culture media similar to physiological fluids. While the use of natural reproductive fluids in culture media may seem impractical at present, the inclusion of some of these proteins produced in the laboratory, such as recombinant albumin, may help to shorten the enormous distance between synthetic media and natural fluids, reducing the stress for the embryo.

To bridge the discrepancy between chemical and biological media, we have performed the first comparative study of the proteomes obtained from three different fluids collected simultaneously from the same donor, namely uterine fluid (UF), oviductal fluid (OF), and plasma (P), aiming to find low abundance proteins of special significance for embryo development. Samples were collected from healthy and fertile women who underwent a salpingectomy. The amount and quality of the samples allowed us to perform immuno-depletion of the high-abundance serum proteins to detect specific lower-abundance proteins in each fluid [22]. The comparison between the three types of fluids and between individuals constitutes the first quantitative comparison of human oviductal and uterine proteins and could help to improve embryo culture media.

## 2. Results

### 2.1. Sample Collection and Immunoaffinity Depletion

OF, UF, and P were collected from three healthy women (BRA52, BRA54, and BRA57) to compare their proteomic profile. The volumes of reproductive fluids obtained from each donor are shown in Table 1.

As for plasma, 500 µL of the sample was obtained in the three cases. Immunoaffinity chromatography-based depletion of the most abundant proteins was performed (Appendix A). The unbound fraction (Mr < 3000 Da), containing the low-abundance protein fraction of each sample, was used for the analysis. The MS and MSMS spectra obtained were used for identification and label-free quantification of the samples, using the statistical design developed by Proteobotics SL. Identified peptides and proteins and their corresponding statistical estimates of significance are described in Appendix A. The parameters used for the search, as well as the codes for the four search engines used, are summarized in Appendix A.

### 2.2. Data Analysis and Quantification

#### 2.2.1. Hierarchical Clustering

Label-free quantification (LFQ) values were analyzed by hierarchical clustering. Samples were grouped into two main clusters: plasma samples and reproductive fluids. The cluster corresponding to reproductive fluids was divided into two subpopulations corresponding to oviductal and uterine samples, respectively, where the samples from patients BRA54 and BRA52 were closer (Figure 1).

#### 2.2.2. Principal Component Analysis (PCA)

PCA showed an increased dispersion of the samples compared to the previous hierarchical clustering. However, it was still possible to distinguish the three groups of samples: OF, UF, and P. The population most closely grouped were oviductal samples, while uterine samples showed the highest dispersion (Figure 2).

#### 2.2.3. Comparative Analysis

Quantitative data obtained for the three groups of samples were normalized and compared: OF versus UF (Appendix A), OF versus P (Appendix A), and UF versus P (Appendix A). Regulated proteins for each paired comparison were color-coded according to the statistical confidence (Table 2).

The comparison between UF and P identified 1602 proteins with at least two unique peptides (false discovery rate FDR <1%) and, thus, was selected for further comparative analysis. Of these, 119 proteins met the criteria established for considering a protein as differentially expressed (*q* < 0.05). In addition, 1625 proteins were identified and quantified when OF was compared to P showing a higher number of either up- or downregulated proteins (*n* = 256) with *q* < 0.05. The lowest number of proteins (*n* = 70, *q* < 0.05) differentially expressed was found in the comparison between reproductive fluids, in which 1626 proteins were identified and quantified.

The pairwise comparison with a higher percentage of proteins differentially expressed (*q* < 0.05) in relation to the total number of proteins identified was the OF-P (15.8%), and the lowest was OF-UF (4.3%) (Figure 3).

In the UF/P comparison, 7.4% of proteins were differentially abundant. This trend remained when low and high abundant proteins were compared. These results confirm both hierarchical and principal component analyses, which revealed the highest differences between OF and P samples. Conversely, both reproductive fluids (OF and UF), showed the lowest percentage of proteins differentially abundant, confirming the highest degree of similarity of these reproductive fluids (Figure 3).

A Venn analysis was performed considering all proteins differentially abundant (*q* < 0.05) (Figure 4), with the main aim of detecting specific differences between low and high abundant proteins both in reproductive fluids and plasma.

When only the low abundant proteins were considered (Figure 4A), it was observed that the highest number of proteins were found when OF was compared to P (96). Additionally, 11 of these low abundant proteins were also found when we compared UF versus P and UF versus OF (Figure 5, Appendix A). STRING network analysis of these 11 proteins showed several biological processes involved in the regulation of protein processing: the protein activation cascade and complement activation. The reactome pathways were mainly from the complement cascade and the innate immune system.

However, there were 39 low abundant proteins in OF compared to P (Figure 6, Appendix A). STRING analysis of these proteins also showed that they were involved in the regulation of protein processing, protein activation cascade, and complement activation. Reactome pathways were mostly related to the complement cascade, the innate immune system, and the regulation of insulin growth factor. In the UF/P comparison, 53 proteins were downregulated, 3 of which were exclusively low abundant in this fluid (retinol-binding protein 4, fibronectin, and flasminogen (Appendix A). Both reproductive fluids (OF and UF) showed only 11 out of 29 proteins significantly low abundant, including lactotransferrin, neutrophil gelatinase-associated lipocalin, and coactosin-like protein, among others (Appendix A). STRING analysis (Figure 6) of these 11 proteins linked reactome pathways from the immune system and signaling by interleukins between most of the proteins, besides Coatomer subunit beta (COPB2), adapter molecule crk isoform Crk-II (CRK), Thy-1 membrane glycoprotein (THY1) and secreted frizzled-related protein 4 (SFRP4).

Both reproductive fluids showed 39 low abundant proteins in comparison to P (Appendix A), mainly linked to regulation of protein processing, protein activation cascade, complement activation, and formation of fibrin clots. The highest number of high abundant proteins was found when OF was compared to P (154), with 92 of these exclusively found in the comparison OF/P (Appendix A). The UF showed 65 high abundant proteins compared to P, with 23 of them exclusive to this comparison (Appendix A). The comparison between both reproductive fluids showed that only 18 out of 40 proteins were exclusively high abundant (Appendix A). Together, the reproductive fluids showed 40 high abundant proteins compared to P (Appendix A) and 2 high abundant proteins (40S ribosomal protein S19 and Tubulin alpha-1C chain) were found in the three comparative pairs: UF versus P, OF versus P, and UF versus OF (Appendix A).

#### 2.2.4. Targeted Mass Spectrometry (MRM/SRM)

Samples (*n* = 9; 3 OF, 3 UF, and 3 P) were reanalyzed by target mass spectrometry to validate 23 proteins selected on the basis of differences found in the quantitative analysis (some were classified as confident high abundant or confident low abundant, others as putative or likely high or low abundant) and based on their previously described roles in fertilization or embryo–maternal communication (e.g., OVGP1. PLMN, HS90A, or PDIA3) [23,24,25,26] (Appendix A).

We confirmed that most of the low abundant proteins (RL3, GSTA1, EZRI, DPYL3, GARS, HSP90A) were significantly present in the three types of fluids according to this sequence: OF > UF > P or OF > UF and absent in P (TSTD1, OVGP1, NNRE, EF2) (Appendix A, Figure 7). However, some proteins (EF1D, PDIA3, CLH1, ENPL) were present in similar amounts in reproductive fluids: OF = UF > P or OF = UF and absent in P (PTGR2, THY1, IPO9, NGAL) (Appendix A, Figure 7). For these proteins (TSTD1, OVGP1, NNRE, EF2, PTGR2, THY1, IPO9, NGAL), we were unable to confirm their presence in plasma samples as we could not detect any of the proteotypic peptides designed as protein-specific targets (Figure 7).

In addition, there were some samples where we detected only one of the proteotypic peptides (e.g., TRFL). On the other hand, SFRP4, COTL1, and TRFL seemed to be more abundant in UF than in the other samples. Finally, some proteins (e.g., TETN, PLMN) appeared to be more abundant in P than in the reproductive fluids.

## 3. Discussion

Different IVF culture media influence the rates of successful implantation, pregnancy, and neonatal outcomes [15]. Recently, it has been shown that culture media supplemented with natural female reproductive fluids has improved IVF efficiency, morphological embryo quality, and epigenetic reprogramming profiles in pig blastocysts, compared to culture media without these supplements [27]. This finding has encouraged the study of the reproductive fluids aimed at the detection of elements that are lacking in embryo culture media [28].

### 3.1. How to Analyze the Reproductive Fluids?

#### 3.1.1. The Most Adequate Method to Collect and Perform A Proteomic Characterization

Despite the oviduct being necessary in nature for optimal gamete maturation, capacitation, selection, and embryo development, detailed information about oviduct secretions and function is still scarce, mainly in humans, due to the difficulty in obtaining appropriate samples [21]. The proteomic characterization of uterine fluid [29] has been recognized for essential components that could be added to the culture media. However, the different collection methods (e.g., aspiration or uterine flushes) have not yet revealed a consistent proteomic pattern of this fluid [30,31,32]. This study shows a proteomic characterization of the low abundance proteins in reproductive fluids (oviductal and uterine) that are significantly detected relative to the plasma of healthy, young, and fertile women during the secretory phase of the menstrual cycle.

The method to collect human reproductive fluids is one of the challenges for the proteomic study of these fluids, mainly due to the small volume of the samples and the difficulty of accessing samples from healthy women without damaging the endometrial cavity or the oviduct. Such damage could result in subsequent bleeding, mostly in the labile endometrium, which could alter the original composition of the collected fluids. In this study, the reproductive fluids were collected with the Mucat device (CDD Laboratoire) adapted with a thinner and smoother distal tip, which reduces the risk of endometrial damage and blood contamination of the samples. This collection method overcomes the limitations to those described previously, such as the reduced volume obtained by aspiration with embryo transfer devices and washing of soluble factors from the glycocalyx with uterine flushes [33]. Our collection method provided enough volume to efficiently perform the protein analysis and quantification by 1D-nano LC ESI-MSMS. The study of the transcriptome and secretome of reproductive fluids achieved through cutting-edge and robust technology used in our work is important to elucidate the role of the female tract and the identification of potentially crucial oviductal factors contributing to the success of fertilization and early embryonic development [16,34,35]. Previous proteomic studies have characterized the greater presence of serum proteins (e.g., serum albumin and immunoglobulins) in these fluids [36], but their abundance can mask the detection of less abundant proteins, which could be of interest in the future development of culture media similar to physiological fluids. Therefore, we decided to perform an efficient depletion of these major proteins to study the reproductive fluids secretome. The MS and MSMS spectra obtained were used for the identification and quantification of the samples, free labeling or label-free format, using the powerful statistical design developed by Proteobotics SL [37], that eliminates likely false positives in peptide and protein identification. Subsequently, a considerable number of proteins connoted as differentially abundant with statistical significance (*q* < 0.05) were identified, some of them which have not been previously identified in other studies. One example of the robust nature of this statistical design is the fact that changes in protein expression identified the oviduct-specific glycoprotein (Q12889) only as a protein putatively high abundant in OF, even though its abundance in this fluid is much higher than in UF or P [38,39]. These data confirm that the proteins detected as differentially abundant under our conditions were accurate. Indeed, with the MSM approach, OVGP1 was detected with higher abundance in OF than in UF and was not detected in plasma, confirming what was expected.

#### 3.1.2. The Most Suitable Study Population

Our set of samples of only three individuals was highly valuable compared to other studies because it was possible to test the reproductive fluids and plasma of the same women, avoiding bias due to individual variability. Another factor that improved efficiency was the homogeneity of the samples, ensured by the strict inclusion criteria that we established: less than 40 years old, healthy, no use of birth control pills, fertile (at least one healthy child born) and collection at the same phase of the menstrual cycle (secretory phase). This phase of menstrual cycle was selected since our goal was to identify low abundant proteins present in these fluids after ovulation and until implantation, since it is the phase corresponding to the time point when human embryos develop under in vitro conditions (from ICSI or IVF until embryo transfer at day 3 or 5) and embryo–maternal communication is highly necessary. This phase occurs after ovulation, when the endometrium experiences several changes, including the transformation of glands and slowing of stromal proliferation [40]. Additionally, it is characterized by the abundant presence of endoplasmic reticulum in the glandular epithelial cells, displacement of nuclei centrally, and accumulation of glycogen-rich vacuoles, which are lost 6 days after ovulation and corresponds to maximal glandular secretory activity [41].

### 3.2. Common Differences in Protein abundance of Reproductive Fluids Versus Plasma

The proteomic analysis performed in this study validates that OF, UF, and P are different from each other (Figure 1 and Figure 2), but the reproductive fluids share most of the identified proteins. In humans, the intramural portion of the uterine tube does not allow a real physical separation between the oviductal and uterine environments. Therefore, it is reasonable to think that there is smooth communication between these anatomical regions. This fact is corroborated by the ability of human embryos, such as other primate embryos, to develop if exposed to the uterine environment prematurely or to implant in the oviduct (ectopic pregnancy), which does not happen in other non-primate mammals. Furthermore, in cases of hydrosalpinx, bilateral salpingectomy is recommended for women before undergoing IVF to improve birth success rates, since hydrosalpinx fluid may alter endometrial receptivity [42,43,44].

Although the importance of prostaglandins in the oviduct has been previously highlighted [45], our study showed for the first time that prostaglandin reductase 2, which forms the stable prostaglandin PGD2, PGE2, or PGF2α, is high abundant in OF in comparison to P, by shotgun and MRM analysis. This analysis also detected Prostaglandin reductase 2 in uterine fluid samples (Appendix A). Elongation factors 2 and 1 were mainly high abundant in OF compared to plasma by the shotgun approach, but MRM analysis also detected significant differences between OF and UF. The reduced expression of ELF1 was proposed as a candidate marker for early diagnosis of cervical cancer [46]. Protein disulfide-isomerase A3 was also detected in the reproductive fluids of these healthy women compared to P by shotgun and MRM analysis, although these proteins were detected before in the endometrium from early-secretory (LH + 2) to mid-secretory phase (LH + 7) in women with unexplained infertility [47]. Although endoplasmin (ENPL) has been linked mainly to oviductal fluid in other species, our study also detected it in uterine fluid [48].

The low abundant proteins were mainly related to complement cascade, regulation of the inflammatory response, and the protein activation cascade. Plasminogen (PLMN) that was detected and quantified for the first time in OF some years ago [49,50] was low abundant in all comparison pairs FO/P, FO/FU, FU/P by shotgun analysis, but MRM analysis did not detect any significant differences. It has been previously shown that the plasminogen/plasmin system is activated during gamete interaction and regulates sperm entry into the oocyte [23,49]. Additionally, it is regulated by progesterone at the transcription level [51]. Therefore, we expected high abundance at the proliferative phase and not at the secretory phase when our samples were collected, as indeed was observed. In the same pathway, tetranectin (CLEC3B), a plasminogen-binding protein belonging to the family of C-type lectins [52], was also detected as a low abundant protein in OF and UF when compared to P in our data.

Previous studies have found specific biomarkers of the secretory phase and window of implantation in human endometria, such as osteopontin (OPN), epidermal growth factor (EGF), Fibronectin, Vitronectin, Secreted Phosphoprotein 1 (SPP1), Laminin, Insulin Like Growth Factor Binding Protein 1 (IGFBP1), Transformin Growth Factor (TFG), Homeobox-leucine zipper protein (Hox10), Interleukin-6 (Il6), Leukemia inhibitory factor (LIF), Bone morphogenetic protein 2 (BMP2), Left-right determination factor 2 (LEFTY2,) Cytosolic phospholipase A2 (CPLA2), Prostaglandin G/H synthase 2 (COX2), and Prostaglandin E2 (PGE) [53,54]. Interestingly, in our study, these biomarkers were not detected since they were not differentially expressed in the reproductive fluids. For example, fibronectin in UF was detected as a putative low abundant protein. However, in OF, it was detected as confident low abundant. On the other hand, laminin (Q16363) was detected as putative upregulated in UF relatively to P, while OF, when compared to P, did not show any significant difference in amount detected but was spotted as putative low abundant compared to UF. The transforming growth factor-beta-induced protein (Q15582) was also detected as putative high abundant in UF and putative low abundant in OF. These discrepancies among these array studies have already been described in previous publications for several genes (*De*) [55], which might be explained by differences in the study design or comparison of mid-secretory phase endometrium to either proliferative or early secretory endometrium.

### 3.3. Differentially abundance Proteins in Oviductal Fluid

The changes in protein abundance observed between reproductive fluids and plasma have allowed us to identify a greater number of differentially abundant proteins in the OF than in the UF compared to P. From these, the high abundant proteins identified were predominantly involved in cellular catabolic process, biosynthesis of amino acids and organic substances, organic and aromatic compounds, and catabolic acid signaling. Some of these proteins have not been detected before in the OF, such as CCT2—T-complex protein 1 subunit beta. However, this protein is associated with capacitation-dependent binding of human spermatozoa to homologous zonae pellucidae, so it is reasonable its detection in the fluid where fertilization takes place [56]. Another protein detected was ACTR3—Actin-related protein 3, whose specific function has not been yet determined. However, this protein is a major constituent of the ARP2/3 complex, known to be involved in the maintenance of the asymmetric (MII) spindle position in mouse oocytes [57]. Arpc1 protein, which is related to spermiation under the regulation of estrogen [58], was also detected in our study. Thiosulfate glutathione sulfurtransferase (TST), which provides the link between the first step in mammalian H2S metabolism performed by the sulfide:quinone oxidoreductase [59], was detected as a highly expressed protein for the first time in this fluid when compared to uterine fluid and plasma samples. These results were also confirmed by MRM analysis.

However, our study also corroborated the presence of interesting proteins previously described in other species [16,21], such as oviduct-specific glycoprotein (OVGP1), which was detected in oviductal fluid in higher amount relatively to uterine fluid.

By shotgun analysis, heat shock protein HSP 90-alpha was high abundant exclusively in OF compared to P, but by MRM analysis, it was detected in UF (although in small quantities). This protein acts like a chaperone of the progesterone receptor (PR) and is essential for the maintenance of its functional activity [60]. Therefore, HSP 90-alpha should contribute to the functionality of progesterone, which is crucial for the early-stage embryo–maternal communication and maintenance of pregnancy [61] and could represent a strong candidate to be present in the culture media, apart from OVGP1. Ezrin, detected in our study, was overexpressed in OF compared to P and UF by shotgun and MRM analysis; likewise, Ezrin was previously detected as a candidate with a role in the final process of oocyte maturation, which occurs in the oviduct and involves zona pellucida hardening [26].

Uba3 was also detected as an high abundant protein in the OF compared to P, which is supported by previous studies, where the corresponding gene expression of Uba3 mRNA in the uterus, ovary, skeletal muscle, and neural tissues was detected, with lower abundance in kidney, intestine, stomach, and liver [62].

The high abundance of some proteins involved in apoptotic processes, such as E3 ubiquitin-protein ligase (HUWE1) and DNA fragmentation factor subunit alpha (DFFA), was also evident in our data. This is not surprising since previous studies have shown the high abundance of several apoptotic proteins in the female reproductive tract [63] and even caspases, which are involved in the development of preimplantation human embryos [64].

### 3.4. Differentially abundant Proteins in Uterine Fluid

As the aforementioned stated, UF showed fewer significantly abundant proteins than OF when compared to P. The detected high abundant proteins are involved in the immune response and granulocyte activation. These results corroborate previous studies that have described the complex interaction between stromal cells and immune cells at the secretory phase of the superficial endometrium, namely the CD56-/CD16+ uterine natural killer (uNK) cells, which plays a role in maternal allorecognition of fetal trophoblasts rather than a cytotoxic role. This “cycling” cell population, which is regulated by regional steroid hormones, local chemokines, and interleukins, increases after ovulation and disappears if there is no pregnancy [40]. One of the proteins that was high abundant in our UF samples was the Thy-1 membrane glycoprotein (THY1). The Thy-1 membrane glycoprotein was high abundant exclusively in UF in relation to P by shotgun, but with MRM analysis, it was not possible to detect the peptides in all samples. This protein has been previously linked to the mechanisms that allow trophoblast cells to fuse with maternal host cells and is imperceptible for maternal immune effectors due to the maternal Thy-I supply to trophoblast somatic hybrids [65], therefore, may play an important role in implantation. The shotgun and MRM analysis showed that the secreted frizzled-related protein 4 (SFRP4) was also detected largely in uterine fluid, as previously described [66]. The lactoferrin estrogen-responsive protein, previously detected in the uterus of mice and rats, was also spotted in uterine fluids by the proteomic techniques performed in this study [67].

The clathrin heavy chain 1 was high abundant exclusively in UF related to P by the shotgun approach, but MRM analysis also detected significant differences between OF and P. Clathrin heavy chain has been shown to be important for viability, embryogenesis, and RNA interference (RNAi) in arthropods, such as *Drosophila melanogaster* and *Metaseiulus occidentalis* [68]. The 60S ribosomal protein L3 is a component of the large subunit of cytoplasmic ribosomes [69]. L ribosomal proteins were detected in reproductive fluids compared to P by shotgun and MRM analysis. The same detection was achieved for the first time by Importin-9, although it was previously linked to Type-I interferons IFN-ε [70], which is constitutively expressed by cells of the reproductive tract [71]. Finally, coactosin-like protein was detected by shotgun and MRM analysis for the first time and was exclusively high abundant in UF in relation to P and OF.

The present data corroborate that reproductive fluids represent an important source of biomarkers with potential interest in the development of better embryo culture media and that each one show particular proteomic profiles according to their roles in the different stages at the beginning of the life cycle (i.e., fertilization, first cleavages, blastocyst implantation). Therefore, it would make sense the design of specific culture media be adapted to the needs of the gametes, zygotes, or embryos at every in vitro culture step in the assisted reproduction laboratories.

## 4. Materials and Methods

### 4.1. Study Population

This research study (internal code 2016_3_6_HCUVA) was approved on 25th of April of 2016 by CEIC (Comité etico de Investigación Clinica) Virgen de la Arrixaca. The sampling was carried out at the Service of Obstetrics and Gynaecology of the University Clinical Hospital Virgen de la Arrixaca in Murcia, Spain. Patients who underwent a planned bilateral salpingectomy by laparoscopy, from June 2016 until June 2018 and fulfilled the inclusion criteria, were invited to participate in the study. Inclusion criteria were the following: premenopausal women with no hormonal treatment during the three months before surgery, normal menstrual cycles, and absence of uterine or tubal pathology as established by transvaginal ultrasound analysis. To select a homogenous population, besides the general inclusion criteria, 3 women (out of 62 initially recruited) were selected based on their similar age (31, 33, and 39 years old), proof of healthy progeny, and because surgery was performed in the secretory phase of the menstrual cycle. Samples (UF and OV, and blood) were stored at the reproductive fluid collection of Biobank-Mur (Biobanco en Red de la Región de Murcia, PT13/0010/0018; PT17/0015/0038, integrated in the Spanish National Biobanks Network, B.000859), until proteomic analysis, following standard operating procedures with approval of the Ethical and Scientific Committees of the hospital, and in accordance with Directive 2004/ 23 /EC of the European Parliament and of the Council of March 31, 2004 concerning human blood and its components, Law 14/2007, of July 3, of Biomedical Research, and Royal Decree of Biobanks 1716/2011, of November 18. Informed consent was obtained from all participants. The patients were identified with specific study codes to protect their confidentiality: BRA52, BRA54, BRA57 (Table 3).

### 4.2. Sample Collection

One fasting blood sample was collected, from the median cubital vein before surgery, to an EDTA-containing tube and centrifuged immediately at 1200× *g* for 15 min at 4 °C to separate the plasma fraction from blood cells. Plasma was aliquoted and frozen at −80 °C until analysis. Uterine fluid was obtained with an adapted Mucat device (Laboratoire CCD, Paris, France) in the surgery room before the procedure. This class I medical device, complying with Directive 93/42/EEC, indicated for direct exocervical or endocervical aspiration and Hühner test, was adapted to be easily introduced into the uterus. Once introduced, aspiration of the fluid was performed with the integrated plunger, which slides up and down when pushed by a flexible acetal resin shaft, without a syringe. Aspirated volumes varied from 59 to 260 µL.

Upon laparoscopic intervention, fallopian tubes were removed, transferred to ice-cold Petri dishes, and dissected. Once dissected, fallopian tubes were clamped in both extremities: A—the one closest to the isthmus (the narrowest part of the tube) and B—the one contiguous to the infundibulum (the widest part of the tube). Afterward, with an ascendant manual mechanical pressure (from the extremity A to B), the oviductal fluid accumulated at the B portion of the ampulla. Then, the clamping scissor that closed this end was opened, and the fluid was aspirated with the Mucat device. Aspirated volumes varied from 49 to 70.9 µL.

Once the fluids were collected in EDTA K2 (1.8 mg/mL) tubes, they were immediately centrifuged at 7000× *g* for 15 min at 4 °C to remove cell debris, and the supernatant was aliquoted and frozen at −80 °C until analysis.

### 4.3. Immunoaffinity Depletion

All samples were diluted 5-fold with 1X dilution buffer (Tris-Buffered Saline (TBS); 10 mM Tris-HCl with 150 mM NaCl, pH 7.4), filtered using a 0.45 µm pore-size spin filter to remove particulate materials, and centrifuged at 9000× *g* for 1 min. To deplete Albumin, IgG, α1-Antitrypsin, IgA, IgM, Transferrin, Haptoglobin, α2-Macroglobulin, Fibrinogen, Complement C3, α1-Acid Glycoprotein (Orosomucoid), HDL (Apolipoproteins A-I and A-II), and LDL (mainly Apolipoprotein B)), an AKTAprime plus FPLC (General Electric, Seoul, South Korea) was used to inject 100–200 µL per sample in a Seppro IgY 14 LC5 column (Sigma–Aldrich, St.Louis, MO, USA) with a constant flow rate of 0.5 mL/min for 20 min, followed by a washing step at a flow rate of 2 mL/min for 3 min. Non bound proteins (depleted fraction) were collected in the flow-through fraction. Due to the large volume of collected fractions, depleted samples were concentrated using Amicon™ Ultra-15 Centrifugal Filter Units (Millipore, MA, USA). Concentrated samples were reconstituted in a chaotropic buffer containing 8 M urea, 2 M thiourea, and 100 mM triethylammonium bicarbonate (TEAB) pH 8.5. Concentrated fractions were stored at −80 °C.

### 4.4. Tryptic Digestion

Samples of 10 µg of each depleted protein were dissolved in 8 M urea, 25 mM ammonium bicarbonate, reduced, and alkylated with iodoacetamide, according to a method previously described (López-Ferrer et al., 2004) [72]. Urea concentration was reduced to 2 M with 25 mM ammonium bicarbonate (final volume 40 µL), and the samples digested overnight at 37 °C with trypsin (Sigma-Aldrich), with a sample/enzyme ratio of 25:1. After digestion, samples were desalted using ZipTip C18 (Merck, MA, USA) [72].

### 4.5. Liquid Chromatography and Mass Spectrometer Analysis

A 2 µg aliquot of each digested sample was subjected to 1D-nano LC ESI-MSMS analysis using a nano liquid chromatography system (Eksigent Technologies nanoLC Ultra 1D plus, SCIEX, Foster City, CA, USA) coupled via a Nanospray III source to a high-speed Triple TOF 5600 mass spectrometer (SCIEX, Foster City, CA, USA). The analytical column used was a silica-based reversed-phase Acquity UPLC M-Class Peptide BEH C18 Column, 75 µm I.D. × 150 mm length, 1.7 µm particle size, and 130 Å pore size (Waters, MA, USA). The trap column was a C18 Acclaim PepMapTM 100 (Thermo Scientific, Waltham, MA, USA), 100 µm × 2 cm, 5 µm particle diameter, 100 Å pore size, switched on-line with the analytical column. The loading pump delivered a solution of 0.1% formic acid in water at 2 µL/min. The nano-pump provided a flowrate of 250 nL/min and was operated under gradient elution conditions. Peptides were separated using a 250 min gradient ranging from 2% to 90% mobile phase B (mobile phase A: 2% acetonitrile, 0.1% formic acid; mobile phase B: 100% acetonitrile, 0.1% formic acid). The injection volume was 5 µL.

Data acquisition was performed with a TripleTOF 5600 System (SCIEX, Foster City, CA). Data were acquired using an ionspray voltage floating (ISVF) 2300 V, curtain gas (CUR) 35, interface heater temperature (IHT) 150 °C, ion source gas 1 (GS1) 25, declustering potential (DP) 100 V. All data were acquired using information-dependent acquisition (IDA) mode with Analyst TF 1.7 software (SCIEX, Framingham, MA, USA). For IDA parameters, 0.25 s MS survey scans (mass range 350–1250 Da) were followed by 35 MS/MS scans of 100ms (mass range 100–1800, total cycle time was 4 s). Switching criteria were ion m/z greater than 350 and smaller than 1250, with a charge state of 2 to 5 and an abundance threshold of more than 90 counts (cps). Former target ions were excluded for 15 s. IDA rolling collision energy (CE) parameters script was used for automatically controlling the CE.

### 4.6. Targeted Mass Spectrometry (MRM/SRM)

To validate the results obtained by 1D-nano LC ESI-MSMS, a total of 2 µg per sample was used for SRM / MRM-directed proteomics using an Eksigent 1D Plus liquid nanocromatograph coupled to a SCIEX 55000 QTRAP quadrupole triple mass spectrometer and using a 60-min gradient. A blank was inserted between samples. A total of 291 transitions were monitored, corresponding to 81 specific peptides of 23 different proteins (Table 4).

As the number of transitions was too high to analyze, using a single method, the method of analysis was converted into two sub-methods (1 and 2) where 12 proteins and their corresponding peptides and transitions were monitored in each. Nine total samples (3 OF, 3 UF, and 3 P) were monitored using both sub-methods, and the raw data files in wiff format were analyzed using the Skyline 4.2 program. The analysis determined the areas corresponding to each transition and peptide. In cases where peptides could not be detected in samples, quantification results are not shown. In cases in which it was possible to monitor more than one peptide per protein, the areas corresponding to the transitions of each peptide and the areas of the peptides of each protein were added to obtain a total summed-area for each protein.

### 4.7. Data Analysis and Quantification

MS/MS spectra in the form of raw data files were processed to mgf format using PeakView^®^ 2.2 Software (SCIEX, Foster City, CA, USA) and mgf files searched using Mascot Server 2.5.1, (London, UK) OMSSA 2.1.9, X!TANDEM 2013.02.01.1, and Myrimatch 2.2.140 against a composite target/decoy database built from the 71,785 sequences of the Homo sapiens reference proteome found at Uniprot (January 2018), together with commonly occurring laboratory contaminants. An initial 35 ppm X!TANDEM search was used to recalibrate the precursor ion mass measurements in all MS/MS spectra. Search engines were then configured to match potential peptide candidates with mass error tolerance of 10 ppm and fragment ion tolerance of 0.02 Da, allowing for up to two missed tryptic cleavage sites and a maximum isotope error (13C) of 1, considering fixed carbamidomethylation of cysteine and variable oxidation of methionine, pyroglutamic acid from glutamine or glutamic acid at the peptide *N*-terminus, and acetylation of the protein *N*-terminus. Score distribution models were used to compute peptide-spectrum match p-values1, and spectra recovered by a false discovery rate (FDR) ≤ 0.01 (peptide-level) filter were selected for label-free quantitative analysis using parent ion intensities. The label-free quantification (LFQ) values of protein identified in nine samples by liquid chromatography-tandem mass spectrometry (LC-MS/MS) were analyzed by hierarchical clustering, principal component analysis (PCA) and Venn Diagram using R Core Team (2013) (R: A language and environment for statistical computing, R Foundation for Statistical Computing, Vienna, Austria), was used to observe if the different values would be grouped in a set of clusters corresponding to each type of sample or patient, in which each cluster would be distinct from each other, and the objects within each cluster would be broadly similar to each other. Protein–protein interaction networks were estimated by STRING (Szklarczyk et al. Nucleic Acids Res. 2015 43(Database issue): D447-52). Differential abundance was quantified using linear models, and statistical significance was measured using *q*-values (FDR). All analyses were conducted using software from Proteobotics (Madrid, Spain) [37,73]. The targeted mass spectrometry (MRM/SRM) data, namely the areas corresponding to each transition and peptide, were transformed to log 2 scale and a paired sample *t*-test was performed to determine whether the mean difference between two sets (OF/P, OF/UF, UF/P) was zero.

The mass spectrometry proteomics data have been deposited to the ProteomeXchange Consortium via the PRIDE [1] partner repository with the dataset identifier PXD015980.

## 5. Conclusions

This study presents a high-throughput analysis of female reproductive tract fluids during the secretory phase of the menstrual cycle, which constitutes a novel contribution to the knowledge of the oviductal and uterine secretomes. The present data corroborate that reproductive fluids represent an important source of biomarkers with potential interest in the development of improved embryo culture media. Currently, recombinant albumin is added to the media because it is considered an essential supplement for embryo development and has been shown to be endocytosed by the embryo from the medium [5], as was also demonstrated for the oviductal OVGP1 [74]. Therefore, it is plausible that the reproductive fluids contain many other proteins with important roles in embryo development itself and can also act as carriers for other embryonic growth factors. Previous studies have focused on UF, but our study demonstrates that OF is also a rich fluid with essential proteins that could be a target to improve fertilization rates and early embryo development if used in the culture media, namely EZRIN, HSP90, or OVGP1. More studies with similar designs and with established standard operating procedures are needed to corroborate our results/hypothesis and find consistent markers in secretions of the female reproductive tract.

## Figures and Tables

**Figure 1 ijms-20-05305-f001:**
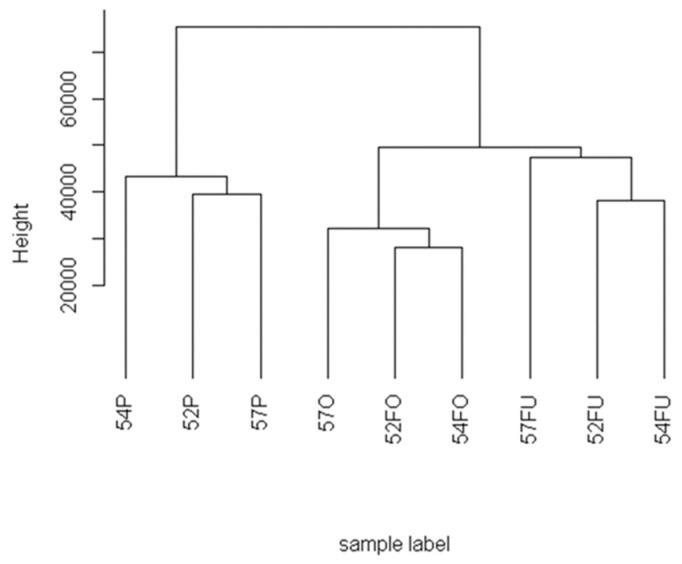
Hierarchical clustering of the samples based on label-free quantification (LFQ) values of protein identified by liquid chromatography-tandem mass spectrometry (LC-MS/MS) of proteolytic peptides from the plasma (P), oviductal fluid (OF), and uterine fluid (UF) samples of the three patients (52, 54, and 57 years old).

**Figure 2 ijms-20-05305-f002:**
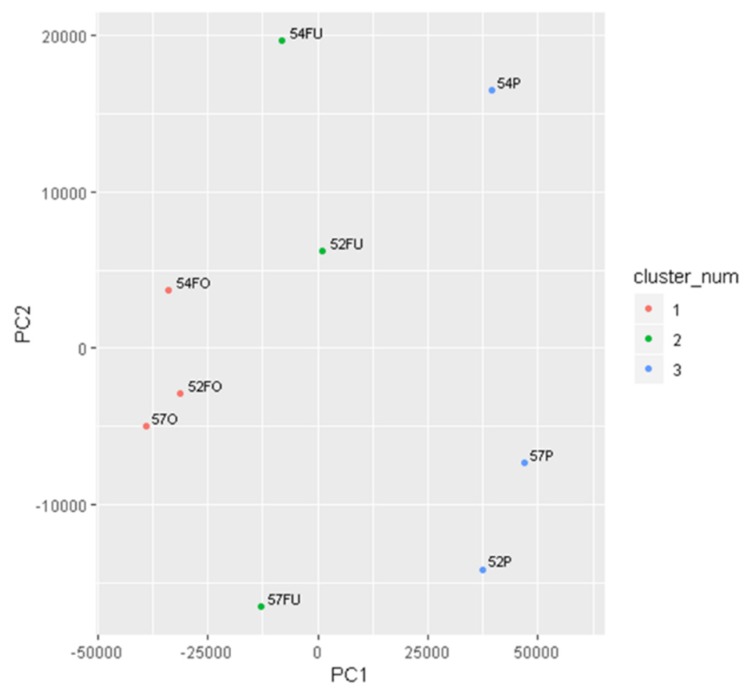
Principal component analysis (PCA) of all identified proteins from label-free experiments.

**Figure 3 ijms-20-05305-f003:**
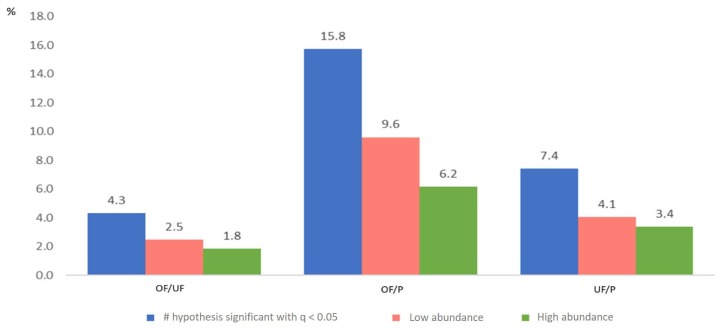
Percentage of proteins differentially abundant after analysis and quantification of 1D-nano LC ESI-MSMS data for three pairwise comparisons: oviductal fluid versus uterine fluid (OF/UF), oviductal fluid versus plasma (OF/P) and uterine fluid versus plasma (UF/P) (*q* < 0.05).

**Figure 4 ijms-20-05305-f004:**
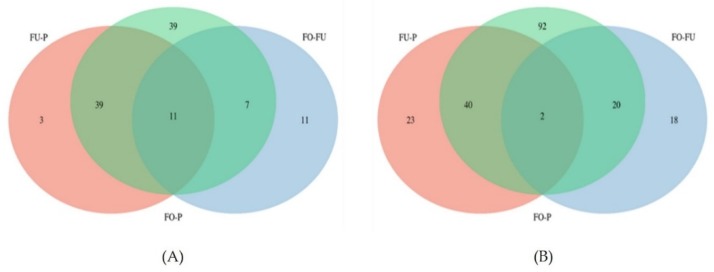
Venn diagram of proteins that showed a significant (*q* < 0.05) low abundance (**A**) or high abundance (**B**).

**Figure 5 ijms-20-05305-f005:**
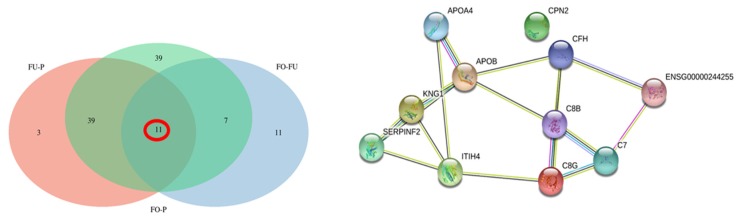
Venn diagram of proteins that showed a significant (*q* < 0.05) low abundance and the corresponding interactome for the 11 low abundant proteins detected in the three comparison pairs (UF/P, OF/P, and OF/UF).

**Figure 6 ijms-20-05305-f006:**
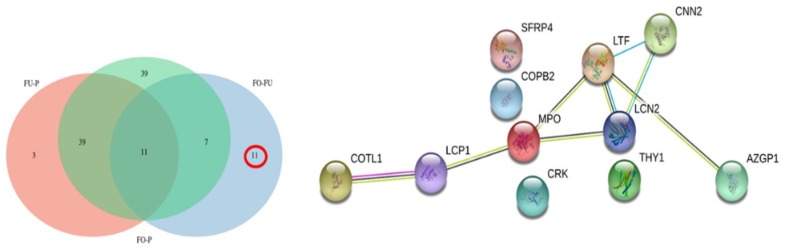
Venn diagram of proteins showing a significant (*q* < 0.05) low abundance and the corresponding interactome for the 11 low abundant proteins in OF compared to UF.

**Figure 7 ijms-20-05305-f007:**
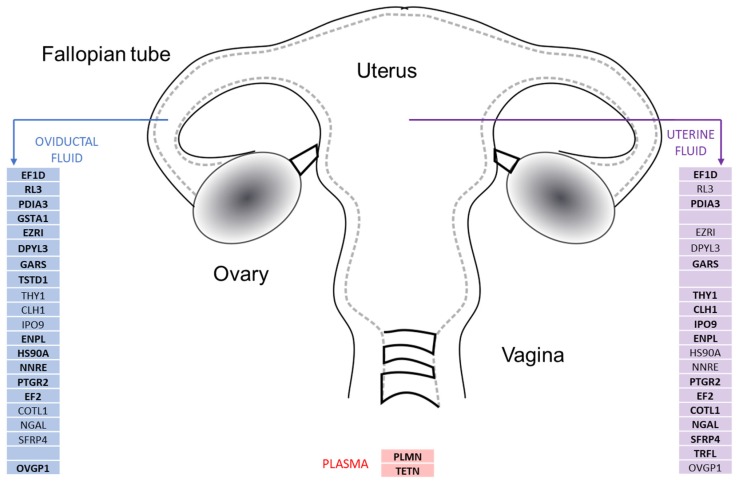
Summary of the low abundance proteins detected by target mass spectrometry, from the 23 selected candidates, in the female reproductive tract fluids. Several proteins were detected in high amounts in reproductive fluids in comparison to plasma samples (EFD1, PDIA3, GARS, ENPL, PTGR2, EF2), while others were low abundant (PLMN, TETN), as expected. Several proteins were detected in high amounts in OF when compared to UF and P samples (RL3, EZRI, DYL3, HS90A, NNRE, OVGP1). On the other hand, the proteins THY1, CLH1, IPO9, COTL1, NGAL, SFRP4 were found in higher amounts in uterine fluid. GSTA1 and TSD1 were detected as exclusively high abundant proteins in oviductal fluid, in comparison to plasma, while TRFL was exclusively high abundant in uterine fluid.

**Table 1 ijms-20-05305-t001:** Volume of reproductive fluids collected from each donor.

Collected Fluids	BRA-52	BRA-54	BRA-57
OF—oviductal fluid (µL)	57	70.9	49
UF—uterine fluid (µL)	260	100	59

**Table 2 ijms-20-05305-t002:** Summary of data analysis and quantification of 1D-nano LC ESI-MSMS for each pairwise comparison (oviductal fluid/uterine fluid (OF/UF), plasma/oviductal fluid (P/OF), and P/UF), indicating the number of abundance proteins for each of the contrasts with different degrees of confidence. The protein abundance for each paired comparison were color-coded according to the statistical confidence (green or red for high grade, yellow or orange for medium grade, yellow or light orange for low degree of confidence and white for no significantly different expression).

OF/UF	OF/P	UF/P	Abundance Color-Code According to the Statistical Confidence
77.90%	60.40%	77.90%	truly null fraction (π0, proportion of contrasts under the null hypothesis)
15	49	13	Confident high abundant (*q*-value < 0.01, positive log fold change)
25	107	52	Likely high abundant (*q*-value < 0.05, positive log fold change)
308	601	398	Putative high abundant (*q*-value > 0.05 but *p*-value < 1−π0, positive log fold change)
1016	552	958	no differentially abundant
232	216	127	Putative low abundant (*q*-value > 0.05 but *p*-value < 1−π0, negative log fold change)
16	33	27	Likely low abundant (*q*-value < 0.05, negative log fold change)
14	67	27	Confident low abundant (*q*-value < 0.01, negative log fold change)
1626	1625	1602	# hypotheses tested
70	256	119	# hypotheses significant with *q* < 0.05

**Table 3 ijms-20-05305-t003:** Demographic data of the recruited patients.

Demographic Data	BRA-52	BRA-54	BRA-57
Age (years)	33	39	31
Menarche (years)	11	14	10
Parity	G4C4	G2P2	G2C1P1
Menstrual cycle duration	30 days	30 days	29/30 days
Sample collection day (Menstrual Cycle phase)	Day 18: early secretory phase	Day 17: early secretory phase	Day 22: secretory phase

**Table 4 ijms-20-05305-t004:** Selected proteins for MRM validation.

Gene Name	Gene Description	Protein Class
EF1D	Elongation factor 1-delta	Plasma proteins Predicted intracellular proteins
RL3	60S ribosomal protein L3	FDA approved drug targets Plasma proteins Predicted intracellular proteins Predicted secreted proteins Ribosomal proteins
PDIA3	Protein disulfide-isomerase A3	Enzymes Plasma proteins Predicted secreted proteins
GSTA1	Glutathione S-transferase A1	Enzymes Plasma proteins Predicted intracellular proteins
EZRI	Ezrin	Cancer-related genes Plasma proteins Predicted intracellular proteins
DPYL3	Isoform LCRMP-4 of Dihydropyrimidinase-related protein 3	Predicted intracellular proteins
GARS	Glycine--tRNA ligase	Disease-related genes Plasma proteins Potential drug targets Predicted secreted proteins
TSTD1	Thiosulfate:glutathione sulfurtransferase	Predicted intracellular proteins
THY1	Thy-1 membrane glycoprotein	CD markers Plasma proteins Predicted membrane proteins Predicted secreted proteins
CLH1	Clathrin heavy chain 1	Cancer-related genes Plasma proteins Predicted intracellular proteins
IPO9	Importin-9	Predicted intracellular proteins Predicted secreted proteins Transporters Predicted localization Intracellular, Secreted
ENPL	Endoplasmin	Cancer-related genes Plasma proteins Predicted intracellular proteins Predicted secreted proteins
HS90A	Heat shock protein HSP 90-alpha	Cancer-related genes Plasma proteins Predicted intracellular proteins
NNRE	NAD(P)H-hydrate epimerase	Disease related genes Enzymes Potential drug targets Predicted secreted proteins
PTGR2	Prostaglandin reductase 2	Enzymes Predicted intracellular proteins
EF2	Elongation factor 2	Cancer-related genes Disease-related genes Plasma proteins Predicted intracellular proteins
COTL1	Coactosin-like protein	Plasma proteins Predicted intracellular proteins
NGAL	Neutrophil gelatinase-associated lipocalin	Candidate cardiovascular disease genes Plasma proteins Predicted secreted proteins
SFRP4	Secreted frizzled-related protein 4	Candidate cardiovascular disease genes Plasma proteins Predicted secreted proteins
TRFL	Lactotransferrin	Cancer-related genes Plasma proteins Predicted intracellular proteins Predicted secreted proteins
PLMN	Plasminogen	Cancer-related genes Candidate cardiovascular disease genes Disease-related genes Enzymes FDA approved drug targets Plasma proteins Predicted secreted proteins
TETN	Tetranectin	Cancer-related genes Plasma proteins Predicted intracellular proteins Predicted secreted proteins
OVGP1	Oviduct-specific glycoprotein	Plasma proteins Predicted secreted proteins

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
