# Peer review of "Which Low-Abundance Proteins are Present in the Human Milieu of Gamete/Embryo Maternal Interaction?"

_ijms, 2019, doi:10.3390/ijms20215305_

Round 1

Reviewer 1 Report

The authors present a manuscript that show a high-throughput analysis of female reproductive tract fluids and contributes to the knowledge of oviductal and uterine secretome. The data corroborate that reproductive fluids represent an important source of biomarkers with potential interest in the development of improved embryo culture media.  Some corrections need to be made before publication on Int. J. Mol Sci.

Introduction

Page 2 line 83: UF, OF and P in full

Results

The results need to be improved.

Page 3 lines 109, 118: figure...Figure

Page 5 lines 143-145: the legend in figure 3 must be correct. “Percentage of proteins diferentially”; “oviductal fluid (OF) versus uterine fluid (UF) (OF/UF), =OF versus plasma (P) (OF/P) and uterine fluid (UF) versus P (UF/P)” ….OF vsUF (OF/UF), OF vsP (OF/P) and UF vsP (UF/P)

Page 7 line 209: figure...Figure

Discussion

Page 9 line 302: table...Table

Page 10 line 351: de...the

Materials and methods

Page 12 line 428: the table number is missing

Page 12 line 446: put the dot and not the comma

Page 12 line 457: write in all the text minutes or min

Page 13 line 493: 2 micrograms…2 mg

 References

Page 18 line 599: “Epigenetic disturbances in in vitro”

Author Response

Thank you for your comments. The suggested corrections have been made in the new version of the manuscript.

Reviewer 2 Report

The present manuscript is very well written and addresses the problematic of human infertility and IVF. The aims and methodologies are well established and present a novelty in the field. Do authors envision to use the data obtained herein to improve the IVF procedures? If so, how do you suggest to do it?

Did authors consider to analyse these fluids during different time points of the menstrual cycle? It could be interesting to analyse the concentration and influence of some biomarkers present in different steps of the menstrual cycle.

Author Response

The present manuscript is very well written and addresses the problematic of human infertility and IVF. The aims and methodologies are well established and present a novelty in the field. Do authors envision to use the data obtained herein to improve the IVF procedures? If so, how do you suggest to do it?

Thank you so much for your kind feedback. We think that further studies must be performed in order to study the impact of the identified proteins in IVF efficiency. However, due the high cost of this strategy of producing each identified protein in vitro we suggest adding directly the fluid to embryo culture, like already was performed in another species ([1][2]). We want to highlight that the oviductal fluid could play an important role on the improvement of IVF procedures because the early embryo development until day 3, under in vivo conditions takes place in oviduct, therefore we strongly believe that the supplement of this fluid will be beneficial.

Cebrian-Serrano, A.; Salvador, I.; García-Roselló, E.; Pericuesta, E.; Pérez-Cerezales, S.; Gutierrez-Adán, A.; Coy, P.; Silvestre, M.A. Effect of the bovine oviductal fluid on in vitro fertilization, development and gene expression of in vitro-produced bovine blastocysts. Reprod. Domest. Anim. 2013, 48, 331–338. Canovas, S.; Ivanova, E.; Romar, R.; García-Martínez, S.; Soriano-Úbeda, C.; García-Vázquez, F.A.; Saadeh, H.; Andrews, S.; Kelsey, G.; Coy, P. DNA methylation and gene expression changes derived from assisted reproductive technologies can be decreased by reproductive fluids. Elife 2017, 6, e23670.

Did authors consider to analyse these fluids during different time points of the menstrual cycle? It could be interesting to analyse the concentration and influence of some biomarkers present in different steps of the menstrual cycle.

Thank you so much for your suggestion. We also collected samples from other time points of the menstrual cycles for further proteomic characterization. Due the high cost of the techniques used to analyze the samples we started to analyze the samples corresponding to implantation, which is extremely important for IVF clinics. We hope to have soon more samples and fund to establish this interesting characterization study.